# Neural Similarity Learning

Weiyang Liu[1,*]    Zhen Liu[2,*]    James M. Rehg[1]    Le Song[1,3]
[1]Georgia Institute of Technology    [2]Mila, Université de Montréal    [3]Ant Financial    [*]Equal Contribution
wyliu@gatech.edu, zhen.liu.2@umontreal.ca, rehg@gatech.edu, lsong@cc.gatech.edu

## Abstract

Inner product-based convolution has been the founding stone of convolutional neural networks (CNNs), enabling end-to-end learning of visual representation. By generalizing inner product with a bilinear matrix, we propose the *neural similarity* which serves as a learnable parametric similarity measure for CNNs. Neural similarity naturally generalizes the convolution and enhances flexibility. Further, we consider the *neural similarity learning* (NSL) in order to learn the neural similarity adaptively from training data. Specifically, we propose two different ways of learning the neural similarity: static NSL and dynamic NSL. Interestingly, dynamic neural similarity makes the CNN become a dynamic inference network. By regularizing the bilinear matrix, NSL can be viewed as learning the shape of kernel and the similarity measure simultaneously. We further justify the effectiveness of NSL with a theoretical viewpoint. Most importantly, NSL shows promising performance in visual recognition and few-shot learning, validating the superiority of NSL over the inner product-based convolution counterparts.

## 1   Introduction

Recent years have witnessed the unprecedented success of convolutional neural networks (CNNs) in supervised learning tasks such as image recognition [20], object detection [47], semantic segmentation [40], etc. As the core of CNN, a standard convolution operator typically contains two components: a learnable template (*i.e.*, kernel) and a similarity measure (*i.e.*, inner product). One active stream of works [13, 63, 25, 61, 8, 53, 24, 59, 26] aims to improve the flexibility of the convolution kernel and increases its receptive field in a data-driven way. Another stream of works [39, 36] focuses on finding a better similarity measure to replace the inner product. However, there still lacks a unified formulation that can take both the shape of kernel and the similarity measure into consideration.

To bridge this gap, we propose the neural similarity learning (NSL) for CNNs. NSL first defines the neural similarity by generalizing the inner product with a parametric bilinear matrix and then learns the neural similarity jointly with the convolution kernels. A graphical comparison between inner product and neural similarity is given in Figure 1. With certain regularities on the neural similarity, NSL can be viewed as learning the shape of the kernel and the similarity measure simultaneously. Based on the neural similarity, we propose the *neural similarity network* (NSN) by stacking convolution layers with neural similarity. We consider two distinct ways to learn the neural similarity in CNN. First, we learn

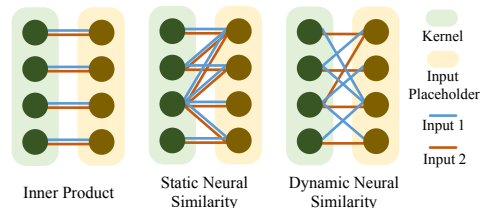

Figure 1: Bipartite graph comparison of inner product, static neural similarity and dynamic neural similarity. A line represents a multiplication operation and a circle denotes an element in a vector. Green color denotes kernel and yellow denotes input.

a static neural similarity which is essentially a (regularized) bilinear similarity. By having more parameters, the static neural similarity becomes a natural generalization of the standard inner product. Second and more interestingly, we also consider to learn the neural similarity in a dynamic fashion.

Specifically, we use an additional neural network module to learn the neural similarity adaptively from the input images. This module is jointly optimized with the CNN via back-propagation. Using the dynamic neural similarity, the CNN becomes a dynamic neural network, because the equivalent weights of the neuron are input-dependent. In a high-level sense, CNNs with dynamic neural similarity share the same spirits with HyperNetworks [18] and dynamic filter networks [28].

A key motivation behind NSL lies in the fact that inner product-based similarity is unlikely to be optimal for every task. Learning the similarity measure adaptively from data can be beneficial in different tasks. A hidden layer with dynamic neural similarity can be viewed as a quadratic function of the input, while a standard hidden layer is a linear function of the input. Therefore, dynamic neural similarity introduces more flexibility from the function approximation perspective.

NSL aims to construct a flexible CNN with strong generalization ability, and we can control the flexibility by imposing different regularizations on the bilinear similarity matrix. In this paper, we mostly consider the block-diagonal matrix with shared blocks as the bilinear similarity matrix in order to reduce the number of parameters. In different applications, we will usually impose domain-specific regularizations. By properly regularizing the bilinear similarity matrix, NSL is able to make better use of the parameters than standard convolutional learning and find a good trade-off between generalization ability and representation flexibility.

NSL is closely connected to a surprising theoretical result in [16] that optimizing an underdetermined quadratic objective over a matrix $W$ with gradient descent on a factorization of this matrix leads to an implicit regularization for the solution (minimum nuclear norm). A more recent theoretical result in [5] further shows that gradient descent for deep matrix factorization tends to give low-rank solutions. Since NSL can be viewed as a form of factorization over the convolution kernel, we argue that such factorization also yields some implicit regularization in gradient-based optimization, which may lead to more generalizable inductive bias. We will give more theoretical insights in the paper.

While showing strong generalization ability in generic visual recognition, NSL is also very effective for few-shot learning due to its better flexibility. Compared to initialization based methods [14, 46], NSL can naturally make full use of the pretrained model for few-shot learning. Specifically, we propose three different learning strategies to perform few-shot recognition. Besides applying both static and dynamic NSL to few-shot recognition, we further propose to meta-learn the neural similarity. Specifically, we adopt the model-agnostic meta learning [14] to learn the bilinear similarity matrix. Using this strategy, NSL can benefit from the generalization ability of both the pretrained model and the meta information [14]. Our results show that NSL can effectively improve the few-shot recognition by a considerable margin.

Our main contributions can be summarized as follows:

- We propose the *neural similarity* which generalizes the inner product via bilinear similarity. Furthermore, we derive the *neural similarity network* by stacking convolution layers with neural similarity. Although this paper mostly discusses CNNs, we note that NSL can easily be applied to fully connected networks and recurrent networks.
- We propose both *static* and *dynamic* learning strategies for the neural similarity. To order to overcome the convergence difficulty of dynamic neural similarity, we propose hyperspherical learning [39] with identity residuals to stablize the training.
- We apply the neural similarity learning to generic visual recognition and few-shot recognition. For few-shot learning, we propose novel usages of NSL and significantly improve the current few-shot learning performance.

## 2  Related Works

**Flexible convolution**. Dilated (atrous) [61, 8] convolution has been proposed in order to construct a convolution kernel with large receptive field for semantic segmentation. [13, 25] improve the convolution kernel for high-level vision tasks by making the shape of kernel learnable and deformable. [39, 36] provide a decoupled view to understand the similarity measure and propose some alternative (learnable) similarity measures. Such decoupled similarity is shown to be useful for improving network generalization and adversarial robustness.

**Dynamic neural networks**. Dynamic neural networks have input-dependent neurons, which makes the network adaptively changing in order to deal with different inputs. HyperNetworks [18] uses a recurrent network to dynamically generate weights for another recurrent network, such that the weights can vary across many timesteps. Dynamic filter networks [28] generates its filters which

are dynamically conditioned on an input. These dynamic neural networks usually perform poorly in image recognition tasks and can not make use of any pretrained models. In contrast, the dynamic NSN performs consistently better than the CNN counterpart, and is able to take advantage of the pretrained models for few-shot learning. [11] investigates the input-dependent networks by dynamically selecting filters, while NSN uses totally different approach to achieve the dynamic inference.

**Meta-learning**. A classic approach [7, 50] for meta-learning is to train a meta-learner that learns to update the parameters of the learner's model. This approach has been adopted to learn deep networks [1, 32, 43, 51]. Recently, There are a series of works [46, 14] that address the meta-learning problem by learning a good network initialization. Specifically for few-shot learning, there are initialization-based methods [43, 46, 14, 10], hallucination-based methods [57, 19, 2] and metric learning-based methods [55, 52, 54]. Besides having very different formulation from the previous works, NSL also combines the advantages from the initialization-based methods and the generalization ability from the pretrained model.

# 3 Neural Similarity Learning

## 3.1 Generalizing Convolution with Bilinear Similarity

We denote a convolution kernel with size $C \times H \times V$ ($C$ for the number of channels, $H$ for the height and $V$ for the width) as $\tilde{W}$. We flatten the kernel in each channel separately and then concatenate them to a vector: $W = \{\tilde{W}^F_{1,:,:}, \tilde{W}^F_{2,:,:}, \cdots, \tilde{W}^F_{C,:,:}\} \in \mathbb{R}^{CHV}$ where $\tilde{W}^F_{i,:,:}$ is the flatten kernel weights of the $i$-th channel. Similarly, we denote an input patch of the same size $C \times H \times V$ as $\tilde{X}$, and its flatten version as $X$. A standard convolution operator uses inner product $W^\top X$ to compute the output feature map in a sliding window fashion. Instead of using the inner product to compute the similarity, we generalize the convolution with a bilinear similarity matrix:

$$f_M(W, X) = W^\top M X \qquad (1)$$

where $M \in \mathbb{R}^{CHV \times CHV}$ denotes the bilinear similarity matrix and is used to parameterize the similarity measure. In fact, if we requires $M$ to be a symmetric positive semi-definite matrix, it shares some similarities with the distance metric learning [60]. Although we do not necessarily need to constrain the matrix $M$, we will still impose some structural constraints on $M$ in order to stablize the training and save parameters in practice. To avoid introducing too many parameters in the generalized convolution operator, we make the bilinear similarity matrix $M$ to be block-diagonal with shared blocks (there are $C$ blocks in total):

$$f_M(W, X) = W^\top \begin{bmatrix} M_s & & \\ & \ddots & \\ & & M_s \end{bmatrix} X \qquad (2)$$

where $M = \mathrm{diag}(M_s, \cdots, M_s)$ and $M_s$ is of size $HV \times HV$. Interestingly, the hyperspherical convolution [39] becomes a special case of this bilinear formulation when $M$ is a diagonal matrix with a normalizing factor $\frac{1}{\|W\|\|X\|}$ being the diagonal. Since additional parameters are introduced to control the similarity measure, we are able to learn a similarity measure directly from data (*i.e.*, static neural similarity) or learn a neural predictor that can estimate such a similarity matrix from the input feature map (*i.e.*, dynamic neural similarity). In the paper, we mainly consider two structures for $M_s$.

**Diagonal/Unconstrained neural similarity**. If we require $M_s$ to be a diagonal matrix, then we end up with the diagonal neural similarity (DNS). DNS is very parameter-efficient and can be viewed as a weighted inner product or an element-wise attention. Besides that, DNS is essentially putting an additional spatial mask over the feature map, so it is semantically meaningful. If no constraint is imposed on $M_s$, then we have the unconstrained neural similarity (UNS) which is very flexible but requires much more parameters.

## 3.2 Learning Static Neural Similarity

We first introduce a static learning strategy for the neural similarity. Specifically, we learn the matrix $M_s$ jointly with the convolution kernel via back-propagation. An intuitive overview for static neural similarity is given in Figure 2(a). When $M_s$ has been jointly learned after training, it will stay fixed in the inference stage. More interestingly, as we can see from Equation (1) that the neural similarity is incorporated into the convolution operator via a linear multiplication, we can compute an equivalent weights for the kernel in advance if the neural similarity is static. Therefore, we can view the new

kernel as $M^\top W$. As a result, when it comes to deployment in practice, the number of parameters used in static NSN is the same as the CNN baseline and the inference speed is also the same.

Learning static neural similarity can be viewed as a factorized learning of neurons. It also shares a lot of similarities with matrix factorization in the sense that the equivalent neuron weights $\hat{W}$ is factorized into into two matrix $M^\top$ and $W$. Although the original weights and the factorized weights are mathematically equivalent, they have different behaviors and properties during gradient-based optimization [16]. Recent theories [16, 5, 33] suggest that an implicit regularization may encourage the

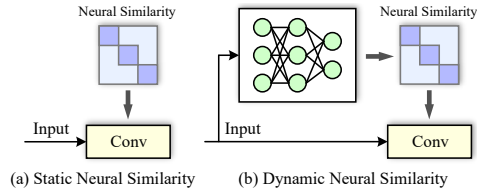

Neural Similarity        Neural Similarity

Input    Conv      Input    Conv

(a) Static Neural Similarity    (b) Dynamic Neural Similarity

Figure 2: Intuitive comparison between static neural similarity and dynamic neural similarity.

gradient-based matrix factorization to give minimum nuclear norm or low-rank solutions. Besides that, we also have structural constraints to explicitly regularize the matrix $M$. Furthermore, we can also view this static neural similarity convolution as a one-hidden-layer linear network. It has been shown that such over-parameterization can be beneficial to the generalization [29, 3, 44, 4].

### 3.3  Learning Dynamic Neural Similarity

#### 3.3.1  Formulation

Besides the static neural similarity, we further propose to learn the neural similarity dynamically. The intuition behind is that the similarity measure should be adaptive to the input in order to achieve optimal flexibility. From a cognitive science perspective, it is also plausible to enable the network with dynamic inference [56, 31]. The difference between static and dynamic neural similarity is shown in Figure 2. Specifically, the dynamic neural similarity is generated dynamically using an additional neural network $M_\theta(\cdot)$ with parameters $\theta$, namely $M_s = M_\theta(X)$. As a result, learning a dynamic neural similarity jointly with the network parameters is to solve the following optimization problem (without loss of generality, we simply use one neuron as an example in the following formulation):

$$\{W, \theta\} = \arg \min_{\{W, \theta\}} \sum_i \mathcal{L}\big(y_i, W^\top M_\theta(X_i) X_i\big) \tag{3}$$

where $y_i$ is the ground truth value for $X_i$, and $\mathcal{L}$ is some loss function. Both $W$ and $\theta$ can be learned end-to-end using back-propagation. Note that, although $X_i$ denotes the entire sample here, $X_i$ will become the local patch of the input feature map in CNNs. For simplicity, we consider a one-neuron fully connected layer instead of a convolution layer. Due to the dynamic neural similarity, the equivalent weights $M_\theta(X)^\top W$ become a function of the input $X$ and therefore construct a dynamic neural network. In fact, dynamic networks which generate the neuron weights entirely based on an additional neural network have poor generalization ability for recognition tasks [18]. In contrast, our dynamic NSN achieves a dedicate balance between generalization and flexibility by using neuron weights that are "semi-generated" (*i.e.*, part of the weights are statically and directly learned from supervisions, and the neural similarity matrix is generated dynamically from the input). Interestingly, we notice that hyperspherical convolution [39] can be viewed as a special case of dynamic neural similarity. One can see that its equivalent similarity matrix $M_\theta(X) = \text{diag}\big(\frac{1}{\|W\|\|X\|}, \cdots, \frac{1}{\|W\|\|X\|}\big)$ also depends on the input feature map but does not have any parameter $\theta$.

**Hyperspherical learning with identity residuals**. In our experiments, we find that naively using a neural network to predict the neural similarity is very unstable during training, leading to difficultly in convergence (it requires a lot of tricks to converge). To address the training stability problem, we propose hyperspherical networks (SphereNet) [39] with identity residuals to serve as the neural similarity predictor. The convergence stability of hyperspherical learning over standard neural networks is discussed in [39, 37, 38, 36, 35, 34]. In order to further stablize the training, we learn the residual of an identity similarity matrix instead of directly learning the entire similarity matrix. Formally, the neural similarity predictor is written as $M_\theta(X) = \text{SphereNet}(X; \theta) + I$ where $I$ is an identity matrix and $\text{SphereNet}(X; \theta)$ denotes the hyperspherical network with parameter $\theta$ and input $X$. To save parameters, we can use hyperspherical convolutional networks instead of hyperspherical fully-connected networks. One advantage of SphereNet is that each element of the output in SphereNet is bounded between $-1$ and $1$ ($[0, 1]$ if using ReLU), making the similarity matrix bounded and well behaved. In contrast, the output is unbounded in a standard neural network, easily making some values of the similarity matrix dominantly large. Most importantly, SphereNet with identity residuals empirically yields not only more stable convergence but also stronger generalization.

### 3.3.2 Disjoint and Shared Parameterization in Neural Similarity Predictor

We mainly consider disjoint and shared parameterizations for the dynamic neural similarity predictor.

**Disjoint parameterization**. Disjoint parameterization treats every dynamic neural similarity independently. For each convolution kernel (*i.e.*, neuron), we use a disjoint neural network to predict the neural similarity matrix $M_s$. A brief overview is given in Figure 3(a).

**Shared parameterization**. Assuming that there exists an intrinsic structure to predict the neural similarity from the input, we consider a shared neural network that produces the neural similarity matrix for different convolution kernels (usually convolution kernels of the same size). To address the dimension mismatch problem of the input feature map, we adopt an adaptation network (*e.g.*, convolution networks or fully-connected networks) to first transform the inputs to the same dimension. Note that, these adaptation networks are not shared across different kernels in general, but we can share those adaptation networks for the input feature map of the same size. An intuitive comparison between disjoint and shared parameterization is given in Figure 3 (Conv1 and Conv2 denote different convolution kernels). By sharing the neural similarity prediction networks across different kernels, the number of parameters used in total can be significantly reduced. Most importantly, this shared neural similarity network may be able to learn some meta-knowledge about the neural similarity.

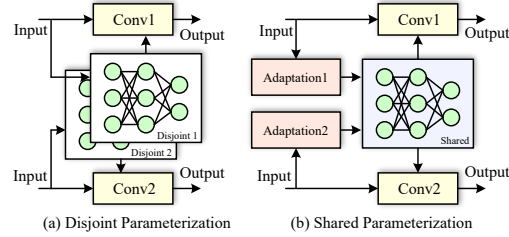

(a) Disjoint Parameterization     (b) Shared Parameterization

Figure 3: Comparison between disjoint and shared parameterization for dynamic neural similarity predictor.

### 3.4 Regularization for Neural Similarity

One of the largest advantages about the neural similarity formulation is that one can impose suitable regularizations on the neural similarity matrix $M$ in different tasks. It gives us a way to incorporate our prior knowledge and problem understandings into the neural networks. The regularization on $M$ controls the flexibility of the neural similarity. If we impose no constraints on $M$, then it will have way too many parameters. Although it may be flexible enough, the generalization is not necessarily good. Instead we usually need to impose some constraints (*e.g.*, the block-diagonal with shared blocks, diagonal, etc.) in order to save parameters and improve generalization.

**Structural regularization**. As a typical example, requiring $M$ to be a block-diagonal matrix with shared blocks is a strong structural regularization. Dilated convolution can be viewed as both structural and sparsity regularization on $M_s$. In fact, more advanced structural regularizations could be considered. For instance, requiring $M$ to be a symmetric or symmetric positive semi-definite matrix is also feasible (by using a Cholesky factorization $M = LL^\top$ where $L$ a learnable lower triangular matrix) and can largely limit the learnable class of similarity measures. Most importantly, structural regularizations may bring more geometric and semantic interpretability.

**Sparsity regularization**. Soft sparsity regularization on the matrix $M_s$ can be enforced via a $\ell_1$-norm penalty. One can also impose a hard sparsity constraint to limit the non-zero values in $M_s$, similar to [42]. It is also appealing to enforce sparsity-one pattern on $M_s$, because it can construct efficient neural networks based on the shift operation in [59].

### 3.5 Joint Learning of Kernel Shape and Similarity

**Formulation**. NSL is also a unified framework for jointly learning the kernel shape and similarity measure. If we further factorize $M_s$ to the multiplication of a diagonal Boolean matrix $D$ and a similarity matrix $R$, then the neural similarity can be parameterized as

$$f_M(W, X) = W^\top \begin{bmatrix} DR & & \\ & \ddots & \\ & & DR \end{bmatrix} X = W^\top \cdot \underbrace{\begin{bmatrix} D & & \\ & \ddots & \\ & & D \end{bmatrix}}_{\text{Kernel Shape}} \cdot \underbrace{\begin{bmatrix} R & & \\ & \ddots & \\ & & R \end{bmatrix}}_{\text{Similarity Measure}} \cdot X \quad (4)$$

where $D = \text{diag}(d_1, \cdots, d_{HV})$ in which $d_i \in \{0, 1\}, \forall i$ is a Boolean value. $D$ actually controls the shape of the kernel because it will spatially mask out some elements in the kernel. Specifically, because the diagonal of $D$ is binary, some elements of $M_s$ will become zero and therefore the kernel shape is controlled by $D$. On the other hand, $R$ still serves as the neural similarity matrix, similar

to the previous $M_s$. $D$ can also be viewed as masking out some elements of each column in $R$. Interestingly, if we do not require the diagonal of $D$ to be Boolean, then it will become a continuous spatial mask for the kernel shape.

**Optimization**. First of all, we only consider $D$ to be static in both static and dynamic NSN. The optimization of $D$ is non-trivial, because it is a Boolean matrix which is discretized and can not be optimized directly using gradients. Therefore, we use a heuristic approach to optimize $D$. Specifically, we preserve a real-valued matrix $D_r$ which is used to construct the Boolean matrix $D$. We define $D = \mathcal{I}(D_r, \alpha)$ where $\mathcal{I}(v, \alpha)$ is an element-wise function that outputs 1 if $v > \alpha$ and 0 otherwise. $\alpha$ is a fixed threshold. We will update $D_r$ with the following equation:

$$\{D_r\}^{t+1} = \{D_r\}^t - \eta \frac{\partial \mathcal{L}}{\partial D} \tag{5}$$

where $D_r$ is only computed in order to update $D$. In both forward and backward passes, only $D$ is used for computation, but $D_r$ is used to generate $D$. Essentially, the gradient w.r.t $D$ serves as a noisy gradient for $D_r$. Similar optimization strategy has also been employed in [22, 12, 42]. $R$ is updated end-to-end using back-propagation. It is also easy to dynamically produce $D$ with a neural network, but we do not consider this case for simplicity.

## 4 Neural Similarity Networks

After introducing the neural similarity learning of a single convolution kernel, we discuss how to construct a neural similarity network using this building block. In order to save parameters, we let all the convolution kernels of the same layer share the same neural similarity matrix, which means that we require the same convolution layer has the same similarity measure. We will empirically validate this design choice in Section 7.1. Stacking convolution layers with static (dynamic) neural similarity gives us static (dynamic) NSN. Note that, static NSN has the same number of parameters as standard CNN in deployment but yields better generalization ability. Compared to [28], dynamic NSN has better regularity on the convolution kernel and is also able to utilize the pretrained CNN models.

**Training from pretrained models**. In order to make use of the pretrained models, we can simply use the pretrained model as our backbone network (with all the weights loaded). Then we add the static or dynamic neural similarity modules to the convolutional kernels and train the neural similarity modules with backbone weights fixed until convergence. Optionally, we can finetune the entire network after the training of the neural similarity module. In contrast, the other dynamic networks [18, 28] are not able to take advantage of the pretrained models. Note that, it is not necessary for both static and dynamic NSN to be trained from pretrained models. They can also be trained from scratch (weights of both backbone and neural similarity module are optimized from random initialization) and still yield better result than the CNN baselines.

**Training and inference**. Similar to CNNs, both static and dynamic NSN can be trained end-to-end using mini-batch stochastic gradient descent. Apart from that the factorized form with $D$ and $R$ needs to be optimized using a heuristic approach, the training is basically the same as the standard CNN. In the inference stage, we can compute all the equivalent weights for static NSN in advance to speed up inference in practice. For dynamic NSN, the inference is also similar to the standard CNN with slightly more additional computations from the neural similarity module.

## 5 Theoretical Insights

### 5.1 Implicit Regularization Induced by NSL

As mentioned before, NSL can be viewed as a form of matrix multiplication where the weight matrix $W$ is factorized as $M^\top W'$ ($W'$ is the new weight matrix and $M$ is the similarity matrix). Such factorization form not only provides more modeling and regularization flexibility, but it also introduces an implicit regularization (in gradient descent). The implicit regularization in matrix factorization is studied in [16]. We first compare the behavior of gradient descent on $W$ and $\{W', M\}$ to observe the difference. We consider a simple example of a one-layer neural network with least square loss (*i.e.*, linear regression): $\min_W \mathcal{L}(W) := \frac{1}{2} \sum_i \|y_i - W^\top X_i\|_2^2$ where $W \in \mathbb{R}^{n \times m}$ is the weight matrix for neurons, $y_i \in \mathbb{R}^m$ is the target and $X_i \in \mathbb{R}^n$ is the $i$-th sample. The behavior of gradient descent with infinitesimally small learning rate can be captured by the differential equation: $\dot{W}_t + \nabla \mathcal{L}(W_t) = 0$ with an initial condition $W_0$, where $\dot{W}_t := \frac{dW_t}{dt}$. For NSL, the objective becomes $\min_{\{W', M\}} \mathcal{L}(W', M) := \frac{1}{2} \sum_i \|y_i - W'^\top M X_i\|_2^2$, so the corresponding differential equations

of gradient descent on $\boldsymbol{W}'$ and $\boldsymbol{M}$ are $\dot{\boldsymbol{W}}'_t + \nabla_{\boldsymbol{W}'}\mathcal{L}(\boldsymbol{W}'_t, \boldsymbol{M}) = 0$ and $\dot{\boldsymbol{M}}_t + \nabla_{\boldsymbol{M}}\mathcal{L}(\boldsymbol{W}'_t, \boldsymbol{M}) = 0$, respectively (with initial condition $\boldsymbol{W}'_0$ and $\boldsymbol{M}_0$). Therefore, the gradient flows of the standard update on $\boldsymbol{W}$ and the factorized NSL update on $\{\boldsymbol{W}', \boldsymbol{M}\}$ can be expressed as

$$\text{Standard Derivative: } \dot{\boldsymbol{W}}_t = \sum_i \boldsymbol{X}_i(\boldsymbol{y}_i - \boldsymbol{W}_t^\top \boldsymbol{X}_i)^\top = \sum_i \boldsymbol{X}_i(\boldsymbol{r}_t^i)^\top \quad (\text{Define } \boldsymbol{r}_t^i = \boldsymbol{y}_i - \boldsymbol{W}_t^\top \boldsymbol{X}_i)$$
$$\text{NSL Derivative: } \dot{\boldsymbol{W}}_t = \boldsymbol{M}_t^\top \dot{\boldsymbol{W}}'_t + \dot{\boldsymbol{M}}_t^\top \boldsymbol{W}'_t = \boldsymbol{M}_t^\top \boldsymbol{M}_t \sum_i \boldsymbol{X}_i(\boldsymbol{r}_t^i)^\top + \sum_i \boldsymbol{X}_i(\boldsymbol{r}_t^i)^\top \boldsymbol{W}'^\top_t \boldsymbol{W}'_t \tag{6}$$

from which we observe that the gradient dynamics of the NSL update is very different from the gradient dynamics of the standard update. Therefore, NSL may introduce a regularization effect that is different from the standard update, and we argue that such implicit regularization induced by NSL is beneficial to the generalization power. [16] conjectures that optimizing matrix factorization with gradient descent implicitly regularizes the solution towards minimum nuclear norm. [5] extends the analysis of implicit regularization to deep matrix factorization (*i.e.*, multi-layer linear neural networks) and shows that multi-layer matrix factorization enhances an implicit tendency towards low-rank solution. [15, 27] show that gradient descent converges to the maximum margin solution in linear neural networks for binary classification of separable data. More interestingly, [5] argues that implicit regularization in matrix factorization may not be captured using simple mathematical norms.

## 5.2 Connection to Dynamical Systems

Classic dynamic neural unit (DNU) [17] receives not only external inputs but also state feedback signals from themselves and other neurons. A general mathematical model of an isolated DNU is given by a differential equation $\dot{\boldsymbol{x}}(t) = -\alpha\boldsymbol{x}(t) + f(\boldsymbol{w}, \boldsymbol{x}(t), \boldsymbol{u})$, $\boldsymbol{y}(t) = g(\boldsymbol{x}(t))$ where $\boldsymbol{x}$ is DNU's neural state, $\boldsymbol{w}_i$ is the weight vector, $\boldsymbol{u}$ is the external input, $f(\cdot)$ is the nonlinear activation and $g(\cdot)$ is DNU's output. As a dynamical system, the output of DNU depends on both the external input and the output time stamp. The neural state trajectory also depends on the equilibrium convergence property of DNU. Different from DNU, dynamic NSN does not have the state feedback and self-recurrence. Instead it realizes the dynamic output with a neural similarity generator that changes the equivalent weight matrix adaptively based on the input. However, it will be interesting to combine self-recurrence to NSL, since it can save parameters and strengthen the approximation power.

Recent work [9, 41, 49, 58] shows that many existing deep neural networks can be consider as different numerical schemes approximating an ordinary differential equation (ODE). NSN with certain similarity design is also equivalent to approximating ODEs. For example, $f_{\boldsymbol{M}} = \boldsymbol{W}^\top(\tilde{\boldsymbol{W}} + \boldsymbol{M})\boldsymbol{X} = \boldsymbol{X}_m + \boldsymbol{W}^\top \boldsymbol{M}\boldsymbol{X}$ where $\boldsymbol{W}^\top\tilde{\boldsymbol{W}} = \text{Diag}(0, \cdots, 0, 1, 0, \cdots, 0)$ (1 lies in the center location) can be written as $\boldsymbol{x}_{n+1} = \boldsymbol{x}_n + \Delta t \cdot g_n(\boldsymbol{x}_n)$ (*i.e.*, ResNet) where $\boldsymbol{x}_n$ is the input feature map at depth $n$ and $g_n(\cdot)$ is the transformation at depth $n$. It is one step of forward Euler discretization of the ODE $\boldsymbol{x}_t = g(\boldsymbol{x}, t)$. Different neural similarity designs correspond to different iterative method for ODEs.

## 6 Discussions

**Connection and comparison to the existing works**. Static NSN is a direct generalization from the standard CNN, and can be viewed as a factorized learning (with optional regularizations) of convolution kernels. Dynamic NSN can be viewed as a non-trivial generalization of hyperspherical convolution [39] in the sense that hyperspherical convolution is also input-dependent and can be viewed as a special case of $\boldsymbol{M}$ being $\frac{1}{\|\boldsymbol{W}\|\|\boldsymbol{X}\|}\boldsymbol{I}$. Compared to dynamic filter networks [28], dynamic NSN achieves a better trade-off between flexiblity and generalization. Dynamic filter networks are very flexible since the weights are completely generated using another network, but it yields unsatisfactory image recognition accuracy. In contrast, dynamic NSN imposes strong regularizations on the weights and is less flexible than dynamic filter networks, but it has much stronger generalization ability while still being dynamic. When $\boldsymbol{M}$ has no constraints, our dynamic NSN will become essentially equivalent to the dynamic filter network. [11] proposes to dynamically select filters to perform inference, while NSL dynamically estimates a similarity measure.

**Dynamic NSN is a high-order function of input**. Dynamic NSN outputs $\boldsymbol{W}^\top \boldsymbol{M}_\theta(\boldsymbol{X})\boldsymbol{X}$. Assuming $\boldsymbol{M}_\theta(\boldsymbol{X})$ is a one-layer neural network, *i.e.*, $\boldsymbol{M}_\theta(\boldsymbol{X}) = \boldsymbol{W}'\boldsymbol{X}^\top$. Then the one-layer dynamic NSN is written as $\boldsymbol{W}^\top \boldsymbol{W}'\boldsymbol{X}^\top \boldsymbol{X}$ which is a quadratic function of $\boldsymbol{X}$. In general, $\boldsymbol{M}_\theta(\boldsymbol{X})$ is much more nonlinear, so one-layer dynamic NSN is naturally a high-order function of the input $\boldsymbol{X}$. Therefore, dynamic NSN has stronger approximation ability and flexibility than the standard convolution.

**Self-attention as a global dynamic neural similarity**. Since self-attention [62] is also a high-order function of input, it can also be viewed as a form of dynamic neural similarity. We define a novel global neural similarity that can reduce to self-attention in Appendix B.

# 7 Applications

## 7.1 Generic Visual Recognition

**Experimental settings**. For fair comparison, the backbone network architecture is the same in each experiment. We will mostly use VGG-like plain CNN architecture. Detailed structures for baselines and NSN are provided in Appendix A. For CIFAR10 and CIFAR100, we follow the same augmentation settings from [21]. For Imagenet 2012 dataset, we mostly follow the settings in [30]. Batch normalization, ReLU, mini-batch 128, and SGD with momentum 0.9 are used as default in all methods. For CIFAR-10 and CIFAR-100, we start momentum SGD with the learning rate 0.1. The learning rate is divided by 10 at 34K, 54K iterations and the training stops at 64K. For ImageNet, the learning rate starts with 0.1, and is divided by 10 at 200K, 375K, 550K iterations (finsihed at 600K).

**Different neural similarity predictor**. We consider two types of architectures: CNN and SphereNet [39] for the neural similarity predictor of dynamic NSN. We experiment on CIFAR-10 and DNS ($M_s$ is diagonal) is used in NSN. Table 1 shows that SphereNet works better than standard CNN as a neural similarity predictor. It is because SphereNet has better convergence properties can can stablize

| Method | Error (%) |
|---|---|
| Baseline CNN | 7.78 |
| Dynamic NSN (CNN) | 7.04 |
| Dynamic NSN (SN) | **6.85** |

Table 1: Predictor Network.

NSN training. In fact, dynamic NSN can not converge if trivially applying CNN to the predictor, and we have to perform normalization (or sigmoid activation) to the predictor's final output to make it converge. In contrast, SphereNet can make dynamic NSN converge easily and perform better. Therefore, we will use SphereNet as the neural similarity predictor for dynamic NSN by default.

**Joint learning of kernel shape and similarity**. We now evaluate how jointly learning kernel shape and similarity can improve NSN. We use CIFAR-10 in the experiment. For both static and dynamic NSN, we use DNS ($M_s$ is a diagonal matrix). For dynamic NSN, we use SphereNet [39] as the neural similarity predictor. Table 2 show that joint learning $D$ and $R$ performs better than simply learning $M_s$. However, to be simple, we will still learn a single $M_s$ in the other experiments.

| Method | Error (%) |
|---|---|
| Baseline CNN | 7.78 |
| Static NSN | 7.15 |
| Static NSN (J) | 6.92 |
| Dynamic NSN | 6.85 |
| Dynamic NSN (J) | **6.64** |

Table 2: Joint learning.

**Shared v.s. disjoint dynamic NSN**. We evaluate the shared and disjoint parameterization for the neural similarity predictor. We use CIFAR-10 in the experiment. For both static and dynamic NSN, we use DNS. Table 3 shows that the shared similarity predictor performs slightly worse than the disjoint one, but the shared one saves nearly half of the parameters used in the disjoint one.

| Method | Error (%) |
|---|---|
| Baseline CNN | 7.78 |
| Dynamic NSN (Shared) | 7.20 |
| Dynamic NSN (Disjoint) | **6.85** |

Table 3: Predictor parameterization.

**CIFAR-10/100**. We comprehensively evaluate both static and dynamic NSN on CIFAR-10 and CIFAR-100. All dynamic NSN variants use SphereNet as neural similarity predictor. Both DNS and UNS are experimented for comparison. Because dynamic NSN uses slightly more parameters than the baseline CNN, we construct a new baseline CNN++ by making the baseline CNN deeper and wider such that the number of parameters is slightly larger than all variants of NSN. The results

| Method | CIFAR-10 | CIFAR-100 |
|---|---|---|
| Baseline CNN | 7.78 | 28.95 |
| Baseline CNN++ | 7.29 | 28.70 |
| Static NSN w/ DNS | 7.15 | 28.35 |
| Static NSN w/ UNS | 7.38 | 28.11 |
| Dynamic NSN w/ DNS | 6.85 | **27.81** |
| Dynamic NSN w/ UNS | **6.5** | 28.02 |

Table 4: Error (%) on CIFAR-10 & CIFAR-100.

in Table 4 verify the superiority of both static and dynamic NSN. Our dynamic NSN outperforms both baseline CNN and baseline CNN++ by a considerable margin. Moreover, one can observe that dynamic NSN performs generally better than static NSN, showing that dynamic inference can be beneficial for the image recognition task. Both DNS and UNS perform similarly on CIFAR-10 and CIFAR-100, indicating that DNS is already flexible enough for the image recognition task.

**ImageNet-2012**. In order to be parameter-efficient, we evaluate the dynamic NSN with DNS on the ImageNet-2012 dataset. The backbone network is a VGG-like 10-layer plain CNN, so the absolute performance is not state-of-the-art. However, the purpose here is to perform apple-to-apple fair comparison. Us-

| Method | Top-1 | Top-5 | # params |
|---|---|---|---|
| Baseline CNN | 42.72 | 19.11 | 8.90M |
| Baseline CNN++ | 42.11 | 18.98 | 9.71M |
| Dynamic NSN w/ DNS | **40.61** | **18.04** | 9.61M |

Table 5: Validation error (%) on ImageNet-2012.

ing the same backbone network, dynamic NSN is significantly and consistently better than both baseline CNN and CNN++. Note that, baseline CNN++ is a deeper and wider version of baseline CNN. The results in Table 5 show that dynamic NSN yields strong generalization ability with the

same number of parameter, and most importantly, the experiments demonstrated that the dynamic inference mechanism can work well in a challenging large-scale image recognition task.

## 7.2 Few-Shot Learning

**Static NSN**. It is very natural to apply static NSN to the few-shot learning. Similar to the finetuning baseline, we first train a backbone network using the base class data. When it comes to the testing stage, we first finetune both the static neural similarity matrix and the classifier on the novel class data and then use the finetuned classifier to make prediction. Note that, in order to use a pretrained backbone, we need to initialize the neural similarity matrix with an identity matrix. Due to the strong regularity that we imposed to the mete-similarity matrix, static NSN is able to preserve rich information from the pretrained model while quickly adapting to the novel class data.

**Dynamic NSN**. Dynamic NSN is very suitable for the few-shot learning due to its dynamic nature. Its filters are conditioned on the input. Because dynamic NSN is able to learn a meta-information about the similarity measure, so its intermediate layers do not need to be finetuned in the testing stage. From a high-level perspective, dynamic NSN shares some similarities with MAML [14] in the sense that dynamic NSN learns to transform its filters with a projection matrix, while MAML transforms its filters using gradient updates during inference. We directly train the dynamic NSN on the base class data. In the testing stage, we first retrain the classifiers using the novel class data, and then classify the query image using the dynamic NSN and the retrained classifier.

**Meta-learned static NSN**. Inspired by MAML [14], we propose to meta-learn the neural similarity. We pretrain the network on the base classes with identity similarity and then meta-learned the neural similarity and classifiers similar to MAML. The meta-learned static NSN dynamically transforms its filters via projection using the gradients, similar to MAML. The meta-optimization is given by

$$\min_{M} \sum_{\tau_i \sim p(\tau)} \mathcal{L}_{\tau_i}(f_{M'}) \quad \text{s.t.} \ \ M' = M - \eta \nabla_M \mathcal{L}_{\tau_i}(f_M) \tag{7}$$

which aims to learn a good initialization for the static neural similarity matrix. During testing, the procedure exactly follows MAML [14] except that the meta-learned static NSN only updates the neural similarity matrix with gradients. The pretrained model is recently shown to perform well with certain normalization [10]. Meta-learned static NSN is able to take full advantage of the pretrained model, and can be viewed as an interpolation between the pretrained model and MAML [14]. In fact, dynamic neural similarity can be also meta-learned similarly, which is left for future investigation.

**Experiment on Mini-ImageNet**. The experimental protocol is the same as [46, 14]. Following [46], we use 4 convolution layers with 32 $3 \times 3$ filters per layer. Batch normalization [23], ReLU non-linearity and $2 \times 2$ pooling are used. For all the NSN variants, we use the best setup and hyperparameters. The results in Table 6 show that all of our proposed three few-shot learning strategies work reasonably well. The dynamic NSN outperforms the other competitive methods by a considerably large margin. Static NSN works better than most exisint methods. Meta-learned static NSN also shows obvious advantages

| Method | Backbone | 5-shot Accuracy |
|---|---|---|
| Finetuning Baseline [46] | CNN-4 | $49.79 \pm 0.79$ |
| Nearest Neightbor Baseline [46] | CNN-4 | $51.04 \pm 0.65$ |
| MatchingNet [46] | CNN-4 | $55.31 \pm 0.73$ |
| ProtoNet [52] | CNN-4 | $68.20 \pm 0.66$ |
| MAML [14] | CNN-4 | $63.15 \pm 0.91$ |
| RelationNet [54] | CNN-4 | $65.32 \pm 0.70$ |
| Static NSN (ours) | CNN-4 | $65.74 \pm 0.68$ |
| Meta-learned static NSN (ours) | CNN-4 | $66.21 \pm 0.69$ |
| Dynamic NSN (ours) | CNN-4 | $\mathbf{71.26 \pm 0.65}$ |
| Discriminative k-shot [6] | ResNet-34 | $73.90 \pm 0.30$ |
| Tadam [45] | ResNet-12 | $76.7 \pm 0.3$ |
| LEO [48] | ResNet-28 | $\mathbf{77.59 \pm 0.12}$ |
| Dynamic NSN (ours) | CNN-9 | $77.44 \pm 0.63$ |

Table 6: Few-shot classification on Mini-Imagenet test set.

over its direct competitor MAML. Moreover, we also compare with the recent state-of-the-art method LEO [48] which uses features from ResNet-28. Our dynamic NSN with the CNN-9 backbone achieves $77.44\%$ accuracy, which is comparable to LEO but ours has much fewer network parameters. This experiment further validates the strong generalization ability of all NSN variants.

## 8 Concluding Remarks

We have proposed a general yet powerful framework to generalize traditional convolution with the *neural similarity*. Our framework can capture the similarity structure that lies in our data of interest, and regularizing the similarity to accommodate the nature of input dataset may yield better performance. Our experiments on image recognition and few-shot learning show the potential of our framework being flexible, generalizable and interpretable. This framework can be further applied to more applications, *e.g.*, semantic segmentation, and may inspire different threads of research.

## Acknowledgements

Weiyang Liu was supported in part by Baidu Fellowship and Nvidia GPU Grant. Le Song was supported in part by NSF grants CDS&E-1900017 D3SC, CCF-1836936 FMitF, IIS-1841351, CAREER IIS-1350983, DARPA Program on Learning with Less Labels.

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
