[Supplementary Material · supplementary.pdf]

# Appendix

## A  Experimental Details

| Layer | CNN-9 (CIFAR-10/100) | CNN-10 (ImageNet-2012) |
|---|---|---|
| Conv1.x | [3×3, 32]×3 | [7×7, 64], Stride 2<br>3×3, Max Pooling, Stride 2<br>[3×3, 64]×3 |
| Pool1 | 2×2 Max Pooling, Stride 2 | |
| Conv2.x | [3×3, 64]×3 | [3×3, 128]×3 |
| Pool2 | 2×2 Max Pooling, Stride 2 | |
| Conv3.x | [3×3, 128]×3 | [3×3, 256]×3 |
| Pool3 | 2×2 Max Pooling, Stride 2 | |
| Fully Connected | 256 | 512 |

Table 7: Our plain CNN architectures with different convolutional layers. Conv1.x, Conv2.x and Conv3.x denote convolution units that may contain multiple convolution layers. E.g., [3×3, 64]×3 denotes 3 cascaded convolution layers with 64 filters of size 3×3.

In CIFAR-10/100, our DNS predictor utilizes the structure of "Input - 64 hidden units (SphereConv, only x being normalized) - 9 output units (no ReLU)", and our UNS predictor uses the structure of "Input - 128 hidden units (SphereConv, only x being normalized) - 81 output units (no ReLU)". Note that, SphereConv comes from [39]. For DNS predictor, we will add an identity matrix to the output of the predictor to improve its initialization point. For UNS predictor, we simply use the output of the network as the neural similarity matrix. For CIFAR-10/100, we use the same training data augmentation as in [36].

On ImageNet-2012 dataset, the DNS predictor uses the structure of "Input - 32 hidden units (Sphere-Conv, only input is normalized) - 9 output units (no ReLU)"

For meta-learning on Mini-ImageNet dataset, we use DNS for all experiments. For our non-MAML baseline and NSN models, we train the models on both training and validation set of Mini-ImageNet, while we train the MAML-trained static NSN model with the training set only.

For non-MAML training, we use Adam optimizer with lr $= 1e - 3$, $\beta_1 = 0.9$, $\beta_2 = 0.999$. For non-MAML testing, we finetune the model on query sets with SGD with lr $= 0.01$, momentum $= 0.9$, dampening $= 0.9$ and weight decay $= 0.001$ for 100 epochs.

In non-MAML static NSN experiments, We train the whole model from scratch and fix the static similarity matrices to be identity; during testing, we only finetune the matrices and the classifier. The (non-MAML) dynamic NSN experiments are similar excluding that we have no static similarity matrix anymore.

In MAML-trained static NSN experiments, we use the trained non-MAML static NSN as a pretrained model, and meta-train both the static similarity matrices and the classifier. For the MAML gradient steps on the support set, we first run 5 gradient steps on both the static similarity matrices and the classifier with step size $= 0.2$. Because MAML-trained static NSN has less capacity for finetuning on query sets, we run additional 20 gradient steps with the same step size but on the classifier only.

The CNN-9 network architecture of dynamic NSN on Mini-ImageNet is the same as the one we use on CIFAR-10/100.

Our code is publicly available at https://github.com/wy1iu/NSL. For all the missing experimental details, please refer to our code repository.

# B  Local and Global Neural Similarity

## B.1  Formulation

The original dynamic neural similarity is performed in a local fashion, meaning that the similarity matrix operates on the local patch instead of the entire input feature map. We extend the original neural similarity from operating on the local patch to operating on the global input feature map. As a result, we call the original neural similarity as *local neural similarity* (LNS). Specifically, for the input feature map $X \in \mathbb{R}^{m \times m \times c}$ with size $m \times m \times c$ and a convolution kernel $W \in \mathbb{R}^{k \times k \times c}$ of size $k \times k \times c$ (stride 1 and dimension-preserving padding), the *global neural similarity* (GNS) for convolution is formulated as

$$F_M^{\mathcal{G}} = W_{\mathcal{G}}^{\top} M_{\mathcal{G}} X_F \tag{8}$$

where $F_M^{\mathcal{G}}$ is a vector of size $mm \times 1$ which is different from the standard neural similarity (with stride 1 and dimension-preserving padding), $W_{\mathcal{G}}$ is the block circulant matrix (a special case of Toeplitz matrix) for performing 2D convolution, $M_{\mathcal{G}}$ is the neural similarity matrix, and $X_F \in \mathbb{R}^{mmc \times 1}$ is flattened vector of the input feature map $X$. The block circulant matrix $W_{\mathcal{G}}^s$ converts the 2D convolution into a matrix multiplication. The GNS matrix $M_{\mathcal{G}} \in \mathbb{R}^{mmc \times mmc}$ usually takes the following block-diagonal form with the same block matrix $M_{\mathcal{G}}^s$:

$$M_{\mathcal{G}} = \begin{bmatrix} M_{\mathcal{G}}^s \in \mathbb{R}^{mm \times mm} & & \\ & \ddots & \\ & & M_{\mathcal{G}}^s \in \mathbb{R}^{mm \times mm} \end{bmatrix} \in \mathbb{R}^{mmc \times mmc} \tag{9}$$

where there are $c$ matrices $M_{\mathcal{G}}^s \in \mathbb{R}^{mm \times mm}$. Note that, if $M_{\mathcal{G}}^s$ is a diagonal matrix, then it will serve as a role similar to a spatial attention mask for the input feature map (The spatial attention map is also shared across different channels of the input feature map if we require $M_{\mathcal{G}}$ to be a block-diagonal matrix with sharing blocks).

Figure 4: Comparison between neural similarity and generalized neural similarity.

**LNS vs. GNS**. The difference between neural similarity and global neural similarity lies in whether the convolution is taken into consideration. For the original neural similarity, although we apply it to convolution kernel, we do not consider the sliding window operation. Instead, we only consider the local inner product operation and combine the neural similarity matrix locally. For global neural similarity, we take the convolution into account and transform the original convolution operation to a matrix multiplication (using Toeplitz matrix). An intuitive comparison is given in Figure 4. From the computation perspective, GNS and LNS are not equivalent in general. For example, we consider the case where both $M$ in LNS and $M_{\mathcal{G}}$ in GNS are diagonal matrix. Although both similarity matrix share the same structure, the equivalent outputs are totally different. For LNS, each position in the output feature map is obtained with a weighted inner product. Diagonal $M$ serves as the element-wise weighting factor for computing the inner product, and the same set of weighting factor will repeatedly be applied to every sliding window (with the same size of convolution kernel) in the input feature map. In contrast, Diagonal $M_{\mathcal{G}}$ in GNS serves as a spatial attention mask for the entire input feature map. It is equivalent to first compute a Hadamard product between the input feature map and the spatial mask induced by $M_{\mathcal{G}}$, and then perform standard 2D convolution with kernel $W$

on the result. GNS and LNS are only equivalent when GNS only considers an input feature map of size $1 \times 1 \times c$ (*i.e.*, the input feature map contains only one spatial location). Both static GNS and dynamic GNS are similar to the corresponding variant in LNS.

**Self-attention as Dynamic GNS**. Dynamic GNS can be written as follows:

$$F_M^{\mathcal{G}} = \boldsymbol{W}_{\mathcal{G}}^\top \cdot \boldsymbol{M}_{\mathcal{G}}(\boldsymbol{X}; \theta) \cdot \boldsymbol{X}_F \tag{10}$$

where $\boldsymbol{M}_{\mathcal{G}}(\boldsymbol{X}; \theta)$ is a function dependent on $\boldsymbol{X}$. We show that self-attention [62] is a special case of dynamic GNS. We first resize the dimension of $\boldsymbol{X}_F$ in Eq. (10) to $mm \times c$ when multiplying with $\boldsymbol{M}_{\mathcal{G}}(\boldsymbol{X}; \theta)$. Then after the multiplication, we resize $\boldsymbol{X}_F$ back to $m \times m \times c$. We consider the case of $\boldsymbol{M}_{\mathcal{G}}(\boldsymbol{X}; \theta) = G_1(\boldsymbol{X}) G_2(\boldsymbol{X})^\top$ where $G_1(\boldsymbol{X})$ is a $1 \times 1$ convolution that transforms $\boldsymbol{X} \in \mathbb{R}^{m \times m \times c}$ to a new feature map with size $m \times m \times c$ and then resize the new feature map to $G_1(\boldsymbol{X}) \in \mathbb{R}^{mm \times c}$. $G_2(\boldsymbol{X})$ is also a combination of $1 \times 1$ convolution and a resize operation, same as $G_1(\boldsymbol{X})$. One can see that $\boldsymbol{M}_{\mathcal{G}}(\boldsymbol{X}; \theta) = G_1(\boldsymbol{X}) G_2(\boldsymbol{X})^\top$ is essentially a self-attention map. By multiplying the self attention map back to the feature map, we have exactly the same self-attention mechanism as in [62]. As a form of dynamic GNS, the self attention operation can be written as

$$F_M^{\text{self-attention}} = \boldsymbol{W}_{\mathcal{G}}^\top \cdot \text{Resize}\left(G_1(\boldsymbol{X}) G_2(\boldsymbol{X})^\top \cdot \text{Resize}(\boldsymbol{X}_F, mm, c), mmc, 1\right) \tag{11}$$

**Connection to spatial transformer**. Dynamic GNS is also closely related to spatial transformer networks [24]. Spatial transformer contains localization network, grid generator, and sampler. In fact, the localization networks take the feature map as input and output parameters for grid generator. Then the grid generator and the sampler transform the feature map. The pipeline resembles the neural similarity learning, and can be viewed as a special case of GNS.

## B.2 Preliminary Experiments

We implement self-attention with our dynamic GNS in both standard CNN and SphereNet [39], and then evaluate them with image classification on CIFAR-10. To simplify evaluation, we only perform mild data augmentation on CIFAR-10 training set, unlike the main paper. We use the CNN-9 architecture in [39] for both standard CNN and SphereNet, but we use 128, 192 and 256 as the number of filters in Conv1.x, Conv2.x and Conv3.x. For more details, refer to our code repository. Table 8 shows the results of CNN and SphereNet with and without self-attention. We can see that self-attention does not seem to bring too many gains to the image classification task. However, we observe that using SphereNet can boost the advantages of self-attention and achieve considerable accuracy gain.

| Method | Accuracy (%) |
|---|---|
| CNN | 90.86 |
| CNN w/ self-attention | 90.69 |
| SphereNet | 91.31 |
| SphereNet w/ self-attention | **91.76** |

Table 8: CNN and SphereNet with self-attention (dynamic GNS) on CIFAR-10.

# C   Significance of NSL for Meta-Learning

One of the key in MAML [14] is that it uses the gradient update to make the network parameter dynamically dependent on the input. Essentially, we can view it as a novel realization of dynamic neural networks except that the network parameters are dynamically changed following the gradient direction. Different from MAML, dynamic NSL realizes the dynamic neural network with an additional neural similarity predictor (*i.e.*, an additional neural network). Essentially, we learn the most suitable direction to update the network parameters adaptively based on the input. As a result, the biggest difference between MAML and dynamic NSL is how we make the network parameters dynamically dependent on the input. MAML uses the gradient information from the gallery set in testing stage, while dynamic NSL learns how to change the network parameters with a neural network during training. Empirically, we find that dynamic NSL outperforms MAML with a significant margin, partially validating that using a neural network to approximate the update of the network parameters yields better generalizability.