[Reviews · NeurIPS 2019]

Reviewer 1



Update: I appreciate the authors taking my feedback seriously. The way the results are presented now is both more complete and easier to interpret. I count on the authors for updating the explanation of shared parametrization (figure and text). ---------- Clarifications/typos: Line 47: inner produce -> product Line 177: not clear to me what the "Shared parametrization scheme" is. What is shared exactly? Is there a single M for the whole network? Line 184: first transform the *inputs* to the same dimension -> are you talking about the images or the feature maps ? Line 380: sifnificant Can you clarify the Shared parametrization strategy ? - What is the input of M(.), the images or the feature maps ? - Is there really just a single M for the whole network ? - When you are resizing "inputs" with the adaptation network, are you talking about feature map or images? Is there just a single adaptation network? - In Fig. 3 there is only one layer so it's not clear how the method behaves for many layers This whole section is very fuzzy; please improve clarity. For the meta-learning experiments, the results can be hard to interpret: * It is well known that for the few-shot learning task, accuracy depends largely on the backbone architecture. * For instance, in the paper "A Closer Look at Few-Shot Classification", the fine-tuning baseline with Resnet-10 architecture achieves 75.90% for miniImageNet 5-shot. * Therefore, it only makes sense to compare meta-learning methods for similar architecture. * The static NSN can be reduced to the classic Conv-4 architecture and can be compared with the other methods. * However, the dynamic NSN cannot really be considered the same architecture, as it has more flexibility than Conv-4.

Reviewer 2



The authors propose to learn a custom similarity metric for CNNs together with adaptive kernel shape. This is formulated via learning a matrix M that modulates the application of a set of kernels W to the input X via f(W, X) = W' M X. Structural constraints can be imposed on M to simplify optimization and minimize the number of parameters, but in its most general form it is capable of completely modifying the behavior of W. Although at test time M can be integrated into the weights W via matrix multiplication, during learning it regularizes training via matrix factorization. In addition, a variant is proposed where M is predicted dynamically given the input to the layer via a dedicated subnetwork. A comprehensive ablation analysis is provided that demonstrates that basic version of the proposed approach performs marginally better than a standard CNN with a comparable number of parameters on CIFAR-10, but the dynamic variant outperforms it by 1%. On ImageNet a 1.5% improvement is demonstrated on the top-1 metric, but a very weak model is used as a backbone (10-layer CNN without batchnorm). Finally, the proposed approach is adapted to the few-shot learning scenario. To this end the basic variant of the model is pretrained on base categories, and the matrices M are fine-tuned on the novel categories with MAML, while the actual CNN kernels remain fixed. This approach outperforms the state-of-the-art LEO method by a statistically significant margin while being much simpler. The paper is well written, and is relatively easy to follow. Overall the approach is interesting but I have several concerns regarding the evaluation (see Improvements). The authors have addressed my concerns as well. Given the results of the additional experiments requested by R1 I'm also ready to recommend the paper for acceptance. However, I would like to point out that, like for most similar approaches, the performance improvements seem to diminish as the network depth increases. In addition, the results in Table 1 in the rebuttal indicate that the meta-learning part of the few-shot learning approach is of a marginal importance. I would appreciate if the authors toned it down in the camera-ready version.

Reviewer 3



#1. The problem tackled in this paper is quite interesting, in which I’ve never seen such work to switch inner product into more general metric. More interestingly, convolutional neural network with generalized inner product with a bilinear matrix is superior to the baseline with the same amount of parameters. #2. I’m very impressed that Dynamic NSN achieves the better few-shot classification accuracy than LEO, even without using residual networks. #3. It's very well-written and easy to follow most of parts in the manuscript. == Updates after the authors’ rebuttal == After reading the rebuttal and having a full discussion, my final recommendation is to accept this paper. Below is a summary of justification to the final score. [Novelty] Though inner product-based convolution is mostly adopted, dynamic neural similarity has some potential to improve the performance of CNN further. Specifically, such generalization seems to be well-suited to few-shot classification, because NSL is theoretically connected to nuclear norm regularization, briefly discussed in the rebuttal. [Experiments] In the response, the authors included some additional experiments on few-shot classification, showing that Dynamic NSN really improves the classification performance on both prototypical network and MAML.

[Author Response · NeurIPS 2019]

We thank all reviewers for the constructive suggestions and the recognition on our novelty. We have carefully addressed every raised concern. We truly hope that the reviewers and AC can reconsider the decision.

**Overall.** We propose a simple and novel neural similarity learning, which enables dynamic inference of CNN. The factorized learning paradigm is shown effective in generic visual recognition, few-shot learning and efficient network construction (in Appendix). NSL is theoretically connected with [Implicit Regularization in Matrix Factorization, NeurIPS 2017] which shows that factorized learning is biased towards the minimum nuclear norm solution. Our factorized learning of neural networks is empirically shown to have inductive bias that generalizes better.

All the reviewers recognize our novelty but concerns about the experimental evaluation and the presentation clarity. We conducted all the requested experiments and will improve the presentation clarity regarding our methodology and implementation. We will also release our implementation code for others to reproduce all the experimental results.

**Reviewer #1. [Input of $M(\cdot)$]** The input of $M(\cdot)$ in the current layer is the feature map from the bottom layer. From another perspective, one can also think that the weights of the entire network depends on the input image, since the weights of every layer recursively depend on the feature map from the bottom layer.

**[Single $M$ for the whole network]** In the dynamic neural similarity, $M$ is dynamic and depends on the input feature map, so for different layer, $M$ will be different (because the feature maps are different). In the shared parametrization scheme, the network that generates $M$ is shared across all the convolution kernels with the same size. Note that, the size of $M$ depends on the size of the convolution kernel.

**[Adaptation modules]** Adaptation modules are used to map the feature map from the bottom layer to a fixed-dimension latent vector, such that all the convolution kernels with the same size can share the neural similarity generation network.

**[Fig. 3 and clarity]** We are very sorry about the confusion in Fig. 3 and we will modify it to reflect multiple layers. For the clarity issue, we will improve the presentation of the detailed implementation in revision.

**[Architecture tweak]** Thanks for the suggestion. We agree with the reviewer that our method can be viewed as an architectural modification which is generally useful. We use a simple way to construct a dynamic network whose equivalent weights are dependent on the input. In fact, meta-learning is just one of the suitable applications for our approach. Moreover, we also showcase the application of NSN in constructing efficient network in Appendix.

**[Additional experiments]** Thanks for the constructive suggestion. We conduct all the meta-learning experiments requested by the reviewer. The results still show very consistent gain introduced by NSN. We will add these new results in revision.

| Method | Finetune | ProtoNet | MAML |
|---|---|---|---|
| Vanilla | 49.79±0.70 / 68.29±0.53 | 68.20±0.66 / 72.26±0.59 | 63.15±0.91 / 67.64±0.85 |
| Static NSN | 66.91±0.66 / 73.63±0.35 | 70.30±0.44 / 75.12±0.37 | 67.87±0.42 / 71.79±0.52 |
| Dynamic NSN | 67.82±0.71 / 78.98±0.60 | 71.39±0.67 / 78.72±0.79 | 68.90±0.65 / 72.23±0.49 |
| Meta-learned static NSN | 69.24±0.69 / 78.05±0.57 | | |

Table 1: 5-shot acc. (%) on Mini-ImageNet. In "a / b", "a" and "b" are the accuracy with CNN-4 and CNN-9, respectively.

**Reviewer #2. [CNN++ in the ablation tables]** The CNN++ in all these ablation tables share the same setting with the "CIFAR-10/100" subsection, so the error rate of CNN++ in all these ablation tables is the same as Table 4, *i.e.*, 7.29%.

**[BatchNorm]** All the baselines and our methods use batch normalization by default, which is also explicitly described in Line 282 of the main paper. (Perhaps the reviewer accidentally missed this?)

| Method | ResNet-18 | ResNet-50 |
|---|---|---|
| Baseline | 31.69 | 24.77 |
| Static NSN | 30.12 | 23.91 |
| Dynamic NSN | 29.68 | 23.45 |

Table 2: Top-1 testing error (%) on ImageNet.

**[Application to ResNets]** Thanks for the suggestion. We test both static NSN and dynamic NSN on ImageNet-2012. To save GPU memory, we use DMS to constrain the similarity matrix $M$ and do not use NSL in 1x1 convolution. Our results show very consistent gain on both ResNet-18 and ResNet-50. We will add these results and related details to revision.

**[Number of parameters]** Dynamic NSN uses ∼25.4M parameters, while wide ResNet-28-10 used in LEO has more than 36.5M parameters. We will add the parameter comparison for all the methods in revision.

**[Deformable convolution (DC)]** DC learns the continuous shape of the kernel, while ours learns both discrete shape and similarity measure jointly. DC show no obvious gain in object classification. We will add discussions in revision.

**[Comparison to TADAM (Oreshkin et al., NIPS'18)]** This is indeed a very related work to compare with. TADAM uses a ResNet-12 as the backbone network, while its additional task embedding network uses two separate fully connected residual networks. The number of the total parameter used in TADAM is already much larger than our method (dynamic NSN), while its 5-shot accuracy on Mini-ImageNet is still 3.2% lower than ours.

**[Data augmentation]** We keep our experiments fair by using the exactly same setting with the prior work. In Table 6, the experimental setting exactly follows the MAML [ICML 2017] paper except for the last two rows. For the last two rows, the setting is the same as the LEO [ICLR 2019] paper. For fairness, We do not perform additional augmentations. Moreover, some methods like LEO perform much worse without data augmentation, so it may be unfair for LEO.

**Reviewer #3.** Thanks so much for the recognition on our work. Please see **Overall.** for a brief discussion about why factorized learning of neural networks leads to better performance.

[Meta-Review · NeurIPS 2019]

The proposed method is very interesting in terms of technical novelty and experiments on few-shot learning. The author response has addressed the reviewers' concerns. Note that the results in Table 1 in the rebuttal indicate that the meta-learning part of the few-shot learning approach is of a marginal importance. So, it would be appreciated if the authors toned it down in the camera-ready version.